# Concomitant Inhibition of FASN and SREBP Provides a Promising Therapy for CTCL

**DOI:** 10.3390/cancers14184491

**Published:** 2022-09-16

**Authors:** Cheng Chi, Lisa Harth, Marina Ramírez Galera, Marina Passos Torrealba, Chella Krishna Vadivel, Carsten Geisler, Charlotte Menné Bonefeld, Pia Rude Nielsen, Michael Bzorek, Jürgen C. Becker, Lise Mette Rahbek Gjerdrum, Niels Ødum, Anders Woetmann

**Affiliations:** 1The LEO Foundation Skin Immunology Research Center, Department of Immunology and Microbiology, Panum Institute, University of Copenhagen, The Maersk Tower, 07.12.76, Blegdamsvej 3C, 2200 Copenhagen, Denmark; 2Department of Pathology, Zealand University Hospital, Roskilde, Sygehusvej 9, 4000 Roskilde, Denmark; 3German Cancer Consortium (DKTK), German Cancer Research Institute (DKFZ), Im Neuenheimer Feld 280, 69120 Heidelberg, Germany; 4Translational Skin Cancer Research, Department of Dermatology, University Medicine, Universitätsstrasse 1, 45141 Essen, Germany; 5Department of Clinical Medicine, University of Copenhagen, 2200 Copenhagen, Denmark

**Keywords:** CTCL, FASN, SREBP, cancer therapy

## Abstract

**Simple Summary:**

The biosynthesis of fatty acids catalysed by FASN plays an important oncogenic role in various malignancies but has not been reported in CTCL yet. Here, we show that FASN is highly expressed in both cell lines and primary cells from CTCL patients. The inhibition of FASN impairs cell viability, survival, and proliferation. FASN expression is partly controlled by SREBP, and dual inhibition of FASN and SREBP enhances the impairment of cell proliferation. Overall, our data suggest that the combination of FASN and SREBP inhibitors could be a promising novel strategy in CTCL therapy.

**Abstract:**

Cutaneous T cell lymphoma (CTCL) is a group of non-Hodgkin’s primary cutaneous T cell lymphomas, with Mycosis Fungoides and Sézary syndrome (SS) being the two most common subtypes. Fatty acid synthase (FASN) is a crucial enzyme that catalyses the biosynthesis of fatty acids, which has been reported to play an oncogenic role in various malignancies but not in CTCL so far. Herein, we show that FASN is highly expressed in CTCL cell lines and in peripheral blood mononuclear cells (PBMCs) from CTCL patients, while it is not in PBMCs from healthy individuals. The inhibition of FASN in CTCL cell lines impairs cell viability, survival, and proliferation, but, interestingly, it also increases FASN expression. However, inhibiting sterol regulatory element binding protein (SREBP), a transcription factor that promotes the expression of FASN, partially reversed the upregulation of FASN induced by FASN inhibitors. Thus, the combination of FASN and SREBP inhibitors enhanced the effects on both CTCL cell lines and PBMCs from SS patients, where a valid inhibition on cell proliferation could be verified. Importantly, compared to non-malignant cells, primary malignant cells are more sensitive to the inhibition of FASN and SREBP, making the combination of FASN and SREBP inhibitors a promising novel therapeutic strategy in CTCL.

## 1. Introduction

Primary cutaneous T cell lymphomas (CTCLs) represent a heterogeneous group of non-Hodgkin’s lymphomas derived from T cells that present in the skin with no evidence of extracutaneous disease at the time of diagnosis. Being the most common type, mycosis fungoides (MF) accounts for 60% of all CTCLs [1]. Another classic type of CTCL Sézary syndrome (SS), defined by peripheral blood involvement, is much rarer but shares many histopathological features with MF [1].

Lipids, including fatty acids (FAs), glycerolipids, glycerophospholipids, sphingolipids, saccharolipids, polyketides, sterol lipids, and prenol lipids [2], are of great importance and play versatile roles in different bioprocesses that may contribute to the occurrence and progression of many diseases. These hydrophobic molecules function as components of organelle and cellular membranes, metabolic substrates and energy storage, regulators, and mediators in cell signalling [3]. Many lipids are synthesized from FAs, and mammalian cells acquire FAs either through endogenous de novo synthesis (DNL) or exogeneous uptake. DNL refers to the biochemical process of synthesizing FAs from acetyl-CoA that are produced most commonly from glucose but also from glutamine and acetate under hypoxia [4,5]. To complete the endogenous formation of FAs, acetyl-CoA is first irreversibly carbonylated into malonyl-CoA by acetyl-CoA carboxylase (ACC), and subsequent condensation of seven malonyl-CoA molecules and one molecule of acetyl-CoA catalysed by fatty acid synthase (FASN) ultimately produces the saturated 16-carbon FA palmitate. Further desaturation and elongation of palmitate give rise to the synthesis of a complex collection of FA-containing lipids. In adults, FASN is distributed mainly in cells with high lipid metabolism such as adipocytes and hepatocytes and in hormone-sensitive tissues such as endometrium, breast, and prostate [6]. Under normal conditions, most cells utilize circulating lipids, while DNL is primarily active in the liver and adipose tissue, where excess carbohydrates can be converted into FAs that are then esterified to storage triacylglycerols (TGs) and provide energy through β-oxidation.

However, metabolic reprograming has become a hallmark of cancer cells, which, unlike normal cells that can acquire sufficient lipids from dietary intake and hepatic DNL, they need extra lipids to sustain tumour survival and growth [7]. In this case, a variety of tumours overexpress FASN, including liver, pancreas, gastric, colorectal, ovarian, breast, and prostate cancer, which is often associated with poor prognosis [8,9,10,11,12,13,14]. The neoplastic deregulation of FASN mainly depends on two pathways; one is posttranscriptional stabilization by ubiquitin specific protease (USP) 2a, and the other is transcriptional upregulation by transcription factor sterol regulatory element binding protein (SREBP) [15,16]. SREBPs are a family of basic-helix-loop-helix-leucine zipper (bHLHLZ) transcription factors with the inactive precursors bound to the endoplasmic reticulum (ER) in the form of the insulin-induced gene (INSIG)-SREBP cleavage-activating protein (SCAP)-SREBP complex. When sterols in the ER decrease, INSIGs are ubiquitylated and thereby degraded, which triggers the SCAP-SREBP complex to escape from the ER and sequentially be transported to the Golgi apparatus via specialized transport vesicles generated by the coatomer complex (COP) II [7]. After consecutive cleavage by membrane-bound transcription factor site 1 protease (MBTPS1) and MBTPS2 in the Golgi, SREBPs release the transcriptionally active NH2-terminal domains, which then enter the nucleus, inducing the expression of a range of genes involved in cholesterogenesis and lipogenesis, including FASN [17]. In humans, there are three isoforms of SREBP, namely SREBP1a, SREBP1c, and SREBP2. Apart from cholesterols, low phosphatidylcholine (PC) levels can also stimulate the proteolytic maturation of SREBP1 through the translocation of MBTPS1 and MBTPS2 from the Golgi to the ER membrane via COPI [18]. Generally, SREBP1c seems more specific to FA synthesis, while SREBP1a and SREBP2 also regulate cholesterol synthesis, but significant overlap exists in target genes of the three isoforms [17].

In tumor cells, aberrant activation and cross talk between multiple signal-transduction pathways that might initially be driven by growth factors and sterol hormones, such as PI3K/AKT and MAPK pathways [16,19,20,21,22,23], stimulate the expression and/or nuclear maturation of SREBP, leading to the amplification of FASN irrespective of extracellular lipid levels [16,17]. By virtue of the distinct lipid metabolism in tumour cells, inhibitors targeting FASN and SREBP have shown great anti-cancer effects in various malignancies [24]. Fatostatin inhibits SREBPs by binding to their escort protein SCAP, thus blocking the ER-Golgi transport of SCAP and preventing SREBP activation [25]. Cerulenin is one of the first compounds found to inhibit FASN by interacting with the FASN β-ketoacyl-synthase domain [26]. C75, which also interacts with the enoyl reductase and the thiosterase domains, is a weaker and irreversible FASN inhibitor designed from Cerulenin to circumvent its chemical instability [27]. Orlistat is an FDA-approved anti-obesity drug that inhibits pancreatic lipase, but it was also found to be a potent inhibitor of FASN via its covalent link to the FASN thiosterase domain [28]. The mode of action of these small molecule inhibitors remains elusive, but mechanisms, including the toxic accumulation of malonyl-CoA, alterations in lipid raft stability, and the induction of ER stress, may relate to apoptosis [29]. Although FASN is widely investigated in solid tumours, only limited numbers of studies focused on lymphomas [30,31,32,33], and to our best knowledge, no studies based on CTCL have been reported.

Here, we investigated the expression and function of FASN in CTCL and found that FASN was highly expressed both in CTCL patients and in CTCL-derived cell lines. In cell lines, inhibiting the function of FASN can reduce cell viability, induce apoptosis, and prevent cell proliferation. These effects can be amplified by the combined inhibition of the transcription factor SREBP due to a further suppression on FASN. In primary cells, despite considerable heterogeneity, SREBP and FASN inhibitors can still decrease overall cellular proliferation, especially in malignant cells compared to non-malignant cells, suggesting a great therapeutic potential of FASN and SREBP in the CTCL.

## 2. Materials and Methods

### 2.1. Reagents and Antibodies

FASN inhibitors Orlistat (#04139), Cerulenin (#C2389), C75 (#C5490), and SREBP inhibitor Fatostatin (#F8932) were all purchased from Sigma Aldrich (St. Louis, MO, USA). Two hundred times diluted TaqMan probes from Life Technologies (Carlsbad, CA, USA) specific for GAPDH (#Hs02758991_g1), FASN (#Hs01005622_m1), SREBP1 (#Hs01088679_g1), SREBP2 (#Hs01081784_m1), and CDKN1A (#Hs00355782_m1) were used for quantitative PCR. Primary antibodies for western blotting targeting β-actine (Santa Cruz Biotechnology, Dallas, TX, USA, AC-15), FASN (Santa Cruz Biotechnology, Dallas, TX, USA, SC-55580), and PARP (Santa Cruz Biotechnology, Dallas, TX, USA, SC-8007) were diluted to 1:1000. Secondary anti-mouse (1:2000, #P0260) and anti-rabbit (1:1000, #P0217) antibodies were purchased from Agilent/Dako, Glostrup, Denmark. Antibodies for flowcytometry included CD3 (145-2C11), CD4 (RM4-5), CD8 (53-6.7), CD7, CD26, Annexin V-FITC, and TCRVβ-PE and were purchased from BD Biosciences (Franklin Lake, NJ, USA).

### 2.2. Cell Culture

All cells were cultured in RPMI-1640 medium (Sigma Aldrich, St. Louis, MO, USA, #R2405) with 1% Penicillin/Streptomycin (Sigma Aldrich, St. Louis, MO, USA, #P7539) at 37 °C with 5% CO_2_. Cell lines used in this study were described elsewhere [34,35,36,37,38]. In brief, Mac1, Mac2a, and PB2B were all from the same patient suffering from lymphomatoid papulosis and anaplastic large cell lymphomas [35]. Mac1 is from circulating tumour cells in the blood sampled during the indolent course of the disease, while Mac2a and PB2B (also called Mac2b) were isolated during a later, more aggressive stage [35]. HH was obtained from the blood of a patient with an aggressive cutaneous T-cell leukaemia/lymphoma [36]. SeAx and SeZ4 were established from the peripheral blood of two different SS patients [37,38]. MyLa2059, Mac1, Mac2a, and PB2B were supplemented with 10% foetal bovine serum (FBS) (In Vitro A/S, Denmark, #BI-04-007-1A). HH cells were supplemented with 20% FBS. SeAx and SeZ4 were supplemented with 10% human serum (HS) (Copenhagen University Hospital Blood Bank, Copenhagen, Denmark) and 1000 U/mL human IL-2 (Novartis, Basel, Switzerland, #004184). PBMCs from SS patients or healthy donors were isolated by Lymphoprep-based (StemCell, Vancouver, BC, Canada, #07861) density-gradient centrifugation and supplemented with 10% HS. Necessary approvals were obtained from the Committee on Health Research ethics (H-16025331) prior to using SS patients’ samples, and the work was performed in accordance with the Declaration of Helsinki.

### 2.3. RNA Purification, Reverse Transcriptase-PCR, and Quantitative PCR

RNA purification via the RNeasy Plus mini kit (Qiagen, Hilden, Germany, #74134) was performed as described before [39], followed by reverse transcriptase quantitative PCR (RT-QPCR) using cDNA reverse transcriptase kits (Applied Biosystems, Waltham, MA, USA, #4368814) according to the manufacturer’s instructions. Quantitative PCR was performed following the guidelines from Life Technologies (Carlsbad, CA, USA), and the amplification and fluorescence detection was conducted in a Lightcycler 480 II instrument.

### 2.4. Protein Extraction and Western Blotting

Protein extraction and western blotting were performed as described earlier [40]. To ensure equal loading, the total protein concentration of each lysate was determined by Bradford assay using Protein Assay Dye Reagent Concentrate (Bio Rad, Hercules, CA, USA, #5000006). The quantification of the bars was conducted on ImageJ.

### 2.5. Immunohistochemical Detection of FASN in Patient Material

Immunohistochemical (IHC) studies were performed on paraffin sections from 19 MF patients. Fourteen patients had early-stage MF (<IIB), and five biopsies were taken from the MF tumor stage (IIB). In brief, staining was performed automatically on Omnis (Agilent/Dako, Glostrup, Denmark), and antigen retrieval was accomplished using EnVision™ FLEX Target Retrieval Solution, High pH (Agilent/Dako, Glostrup, Denmark, GV804). After pre-treatment, slides were incubated with anti-FASN clone C20G5 (Cell Signaling, Danvers, MA, USA, #3180, 1:10). The reactions were detected using the polymer technique EnVision Flex+ (Agilent/Dako, Glostrup, Denmark, GV800+809) and visualized using EnVision™ Flex DAB+ Chromogen system (Agilent/Dako, Glostrup, Denmark, GV825) following the instructions given by the manufacturers. Finally, sections were counterstained with Hematoxylin and mounted with pertex. Controls were xenograft tumour MyLa2059 and two healthy skin biopsies. FASN protein expression (cytoplasmatic and membranous) was scored semi-quantitatively as positively stained lymphocytes (0 = no positive cells; 1 ≤ 10% positive cells; 2 = 10–50% positive cells; 3 ≥ 50% positive cells). Internal positive controls were eccrine glandular and follicular epithelium, as well as epidermis.

### 2.6. MTT Assay

Samples were pre-incubated with an FASN and/or SREBP inhibitor in a flat-bottomed 96-well plate for 48/72 h before adding in MTT buffer (5 mg/mL in PBS (phosphate buffered saline), Sigma Aldrich, St. Louis, MI, USA, #M5655-1G). After 2–4 h of incubation, MTT lysis buffer (195 g SDS, 300 mL DMF (N-demethylformamide), 5 mL 2.5 N HCl, 23 mL 100% acetic acid, and 240 mL H_2_O)) was added, and the plate was incubated overnight at 37 °C to dissolve formazan crystals. Absorbance at 570 nm was measured by a spectrophotometer (Thermo Fisher Scientific Multiskan FC, Waltham, MA, USA).

### 2.7. Flow Cytometry

For cell lines, cells were incubated with FASN/SREBP inhibitors for 72 h before harvesting. For PBMCs from SS patients, three CD3/CD28 beads per 5 cells were added on Day 1 to sustain the cell viability. On Day 3, FASN/SREBP inhibitors were added and incubated for another 72 h. Surface markers were stained in fluorescence-activated cell sorting (FACS) buffer for 30 min in the dark. CellTrace Violet (#C34557) for cell proliferation analysis, Annexin V-PE (#A35111) for apoptosis assay, and MitoTracker (#M7512) and Near-IR(#L34993) for cell viability determination were all purchased from Thermo Fisher Scientific (Waltham, MA, USA). Finally, all samples were analysed on a BD LSRFortessa, and data were analysed using the FlowJo 8 or 10 (TreeStar) software. Relative CellTrace Violet (CTV) level was the multiple of mean fluorescence intensity (MFI) of CTV in experiment groups compared to the untreated control, whose MFI was referred as one.

### 2.8. Statistical Analysis

Statistical analyses were carried out using the GraphPad Prism (Prism for macOS, version 9.4.1, GraphPad Software Inc., La Jolla, CA, USA). The mean ± standard error of mean (SEM) was calculated for each treatment group. Statistical analyses were performed using either the one-way ANOVA test with Tukey’s multiple comparison, or the two-sided Student’s *t* test with a 5% significance level, unpaired observations, and equal variance. Significance was defined as * *p* ≤ 0.05; ** *p* ≤ 0.01; *** *p* ≤ 0.001, **** *p* ≤ 0.0001.

## 3. Results

To determine the expression level of FASN, we analysed the cell lines by qRT-PCR (qPCR) and Western blotting. Despite different origins, all the CTCL cell lines had increased FASN expression compared to healthy PBMC at both the protein and mRNA level (Figure 1 A–C). In addition, 18 of 19 skin biopsies from CTCL patients revealed FASN positivity of a varying degree immunohistochemically, whereas control biopsies only showed FASN staining in adnexa (Figure 1D).

We found a tendency for a higher expression of FASN with increasing clinical stage, but the number of cases enrolled in this study was too small to make any firm conclusions (Table 1).

To determine whether FASN affected cell viability, we blocked FASN function using different FASN inhibitors (Orlistat, Cerulenin, or C75). The cells were incubated with the respective inhibitors for 48 h, and their viability was subsequently measured in MTT assays. The inhibitors all impaired the viability in the different CTCL cell lines tested in a dose-dependent manner. Interestingly, the cell lines had varying sensibility towards C75 (Figure 2C), while they shared the same sensibility to Orlistat and Cerulenin (Figure 2A,B). PBMCs from healthy individuals were irresponsive to Orlistat but responded to Cerulenin as the CTCL lines did (Figure 2A,B). A table listing the IC_50_ values for all three FASN inhibitors can be found in Appendix A.

Interestingly, FASN mRNA and protein in most CTCL cell lines were increased after being treated with Orlistat (Figure 3A,B), indicating a compensatory auto-upregulation in these cells, except in SeAx cells where FASN protein was reduced, while the FASN mRNA level seemed unaffected. The treatment of PBMCs from healthy donors did not affect FASN mRNA expression (Appendix A). Given that one of the major FASN regulators SREBP, which promotes the expression of FASN at the transcription level, can be activated by low lipid levels [18], and studies have shown that the loss of FASN can promote nuclear localization and activation of SREBP2 [41], we hypothesized that one possible explanation could be that dysfunctional FASN due to inhibition led to a decline in FAs and their derivates.

This in turn triggered the upregulation of SREBP and subsequent FASN as a feedback loop to compensate the lipid needs in the cells. Therefore, to verify whether additional inhibition of SREBP could incur further inhibition of FASN or prevent the upregulation of FASN induced by FASN inhibitors, we introduced Fatostatin, an SREBP inhibitor, that represses the activation of SREBP by interfering with the ER-to-Golgi transport of SCAP, and thereby theoretically blunts the feedback loop between FASN expression and cellular lipid level [42].

Quantification of western blot in Figure 3D–F confirmed this hypothesis. Although no obvious changes could be seen in MyLa2059 cells, the combination of Fatostatin and Orlistat led to more reduction of FASN in SeAx cells and partially counteracted the compensatory increase of FASN in SeZ4 cells. A further MTT assay showed that the inhibition of SREBP was sufficient to weaken cell viability (Figure 3C), but when combined with FASN inhibitors, this effect was more potent compared to either of these inhibitors alone in most cases (Figure 3G–J).

Meanwhile, the proportion of live cells evaluated by flow cytometry elucidated that cell survival in CTCL cell lines was also abated by the SREBP inhibitor Fatostatin or FASN inhibitor Cerulenin and C75, and the effect was more prominent when simultaneously blocking the two targets (Figure 4A–C). In addition, apoptosis measured by flow cytometry (Figure 4D–F) and the cleavage of poly (ADP-ribose) polymerase (PARP) examined via western blot (Figure 4G) indicated that cell death induced by inhibitors of SREBP and FASN was at least partially mediated through the induction of apoptosis in SeAx and SeZ4 cells. CellTrace violet (CTV) is a dye that can be incorporated into cells and equally passed on to daughter cells after division, and the more the cells divide, the less the intensity of CTV is. Therefore, to comprehend the influence of SREBP and FASN inhibitors on cell proliferation, we used CTV as the proliferation indicator. In Figure 5A–D, we found a retarded CTV peak and increased mean fluorescence intensity (MFI) in CTCL cell lines, implicating that the simultaneous inhibition of SREBP and FASN prevented cell proliferation with even higher efficacy compared to cells where either SREBP or FASN were singly inhibited. The inhibition of FASN has previously been shown to induce the increased expression of CDKN1A in human colon carcinoma cells [43]. Given that cyclin-dependent kinase inhibitor 1A (CDKN1A) is a common molecule involved in the cell cycle arrest at the G1 checkpoint [44], we next analysed the CDKN1A mRNA level and found an upregulation of CDKN1A in cells treated simultaneously with SREBP and FASN inhibitors, indicating that CDKN1A is involved in cell proliferation suppression (Figure 5E–H).

PBMCs from primary SS patients express high levels of FASN mRNA (Appendix A). Therefore, we next analysed the effect of SREBP and FASN inhibitors on PBMCs isolated from the blood of SS patients. We previously described how to identify the malignant clone in SS patients based on the analysis of TCRvb expression [45]. In SS patient 22 (P22), we found that SREBP and FASN inhibitors restricted cell proliferation, especially when both targets were co-inhibited, with malignant cells (defined as CD3^+^CD4^+^CD8^-^TCRvb^+^) being more sensitive than non-malignant CD4^+^ T cells (defined as CD3^+^CD4^+^CD8^-^TCRvb^-^) (Figure 6A–C). Although these inhibitors may not have remarkable effects on cell survival or apoptosis in primary cells (Figure 6D–G), and the effect of SREBP and FASN inhibitors varied among cells from different patients (Figure 6H–M), proliferation data from other patients still resembled that from P22 (Figure 6H,K). A similar tendency could also be concluded when analysed on CTV raw intensity values instead of relative level (Appendix A), corroborating the cytostatic effect of SREBP and FASN inhibitors.

## 4. Discussion

In tumour cells, to fuel cell growth and division, metabolic reprograming such as the Warburg effect, where energy metabolism is limited to glycolysis even in the presence of oxygen, becomes a hallmark of cancer [46,47]. Another emerging focus of tumorigenesis in the last decades is the insight into fatty acid metabolism since 1953, when DNL was first described in neoplastic tissues [48]. FAs are the main components of the cellular lipid pool that constitute a diversity of lipid species, consequently participating in the synthesis of biological membranes and the modulation of their fluidity, serving as a source of energy and functioning as secondary messengers in signalling pathways. Aside from increased exogenous uptake, DNL in cancer cells allows for more diversity and flexibility in lipid production [49]. Not surprisingly, as the key enzyme in the final step of DNL, anomalous FASN expression contributes to the rewriting of lipid metabolism in many malignancies.

In this study, we showed that FASN was highly expressed in CTCL cell lines (Figure 1A,B), PBMCs from SS patients, and biopsies from CTCL patients, but it was not detected in healthy controls (Figure 1C). The inhibition of FASN by classic FASN inhibitors Orlistat, Cerulenin, or C75 caused a significant drop in cell viability in different CTCL cell lines (Figure 2A–C), while in healthy PBMCs, Orlistat had no effect (Figure 1A). The unresponsiveness of healthy PBMCs to Orlistat is conceivable since FASN is mainly restrained in liver and adipose tissue [50], and even in these lipogenic tissues, the expression and function of FASN were negligible under usual dietary conditions [51]. In support of this, we only found negligible levels of FASN protein expression in PBMCs from healthy donors (Figure 1C). In T cells in particular, lipid metabolism plays a crucial role in determining cell differentiation and function [52,53], and de novo fatty acid synthesis mostly happens in activated T cells [54]. However, a considerable loss of viability in healthy PBMCs treated with Cerulenin and C75 (Figure 2B,C) was also observed, which might be related to other off-target effects caused by nonspecific ring-opening reactions because of the reactive epoxide ring in Cerulenin and C75 [55,56].

Interestingly, despite the potent impact of FASN inhibitors on cell viability, we found that Orlistat elicited an increase in FASN expression, which led us to hypothesize that this could be triggered by a feedback mechanism because of high metabolic flexibilities in tumour cells [57]. Considering that SREBPs are pivotal transcription factors in lipid homeostasis, which oneself can also be regulated by lipid abundance, the SREBP inhibitor Fatostatin was selected to contend against the upregulation of FASN induced by FASN inhibitors. Concomitant inhibition of SREBP and FASN proved that Fatostatin could reduce FASN upregulation caused by Orlistat in SeZ4 cells and almost eliminated FASN protein in SeAx cells (Figure 4G). However, the genuine mechanism of how Fatostatin reduced FASN expression needs further study. In fact, the upregulation of FASN was correlated to the upregulation of both SREBP1 and SREBP2, but the inactivation of SREBP did not impede FASN at the mRNA level (Appendix A). On the contrary, it induced even more transcription of FASN in Mac1 and SeAx cells (Appendix A), indicating that additional posttranslational regulatory mechanisms might exist. Anyhow, compared to the mere inhibition of either FASN or SREBP, the combination of FASN inhibitors and SREBP inhibitors in CTCL cell lines indeed brought in more potency on cell viability and cell survival partially by virtue of apoptosis and concurrently more retardancy on cell proliferation via the upregulation of CDKN1A.

In primary PBMCs isolated from SS patients, it is more difficult to obtain results with statistical significance due to the rarity of CTCL patient samples and high heterogeneity from patients to patients. Nonetheless, the hindrance of cell proliferation by the inhibition of FASN and SREBP appeared in every patient included in this study (Figure 6A). Of note, malignant cells seemed more sensitive to FASN and SREBP inhibitors than non-malignant cells, which is a big advantage as a therapeutic target, despite the cytocidal effect being less profound than in CTCL cell lines. Actually, there are many preclinical and clinical trials in different types of malignancies targeting FASN, but side effects such as drastic weight loss, anorexia, and ineffectiveness in some types of tumours remain a concern [49,58]. In this case, increased antitumor efficiency with no extra harm to non-malignant cells makes the combination of FASN inhibitor and SREBP inhibitor a promising therapy for CTCL.

Overall, in this study, we confirmed the overexpression of FASN in CTCL, preliminarily disclosed its oncogenic role in CTCL, and proposed that the combination of FASN and SREBP inhibitors could be a potential therapy for CTCL patients.

## 5. Conclusions

In this study, we provided the first evidence that FASN is expressed in CTCL cell lines and in primary malignant T cells from SS patients. The inhibition of FASN resulted in reduced proliferation and the survival of malignant cells. We also showed that the expression of FASN is regulated by SREBP1/SPREBP2 and that the combination of FASN and SREBP inhibitors enhanced the effects on both CTCL cell lines and primary SS patient cells. Taken together, these results suggest that FASN/SREBP could be a novel therapeutic target in CTCL.

## Figures and Tables

**Figure 1 cancers-14-04491-f001:**
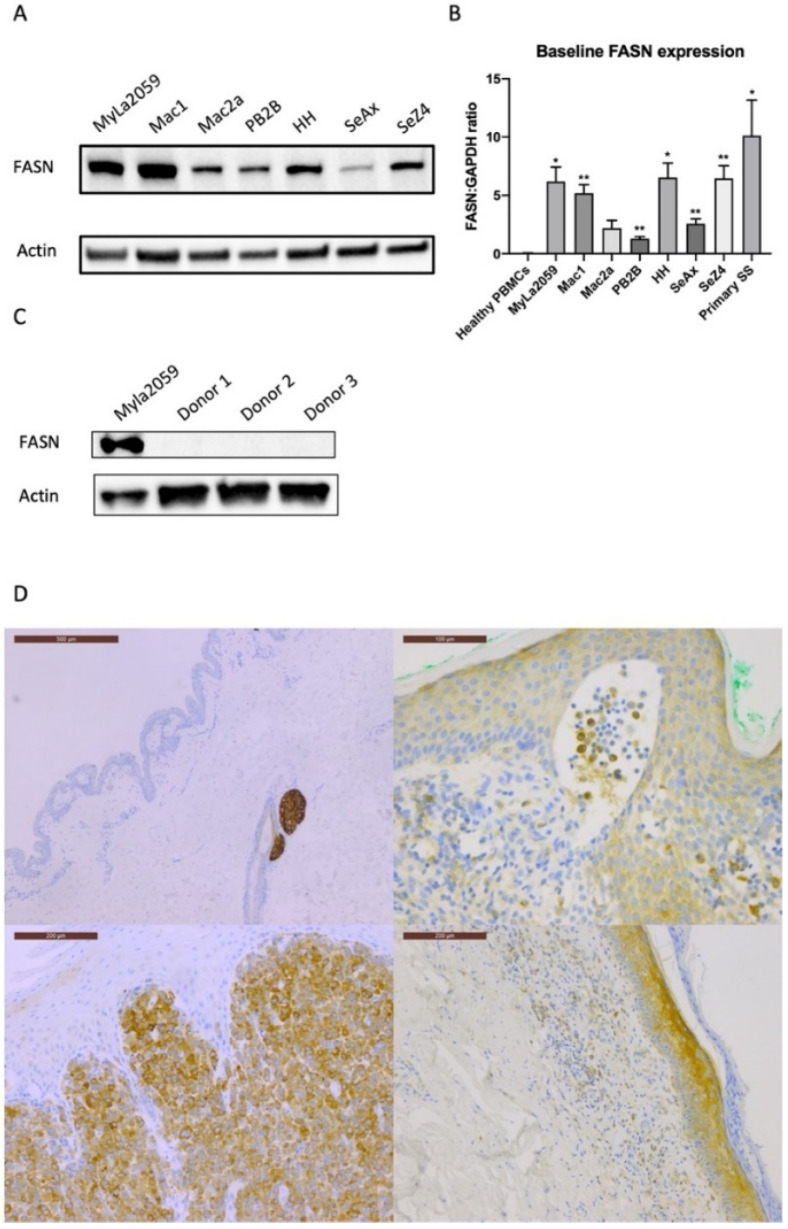
FASN expression in CTCL cell lines and biopsies. (**A**) Western blot and (**B**) qPCR showed increased levels of FASN expression in CTCL cell lines. (**C**) Protein expressions of FASN in PBMCs from three healthy donors were compared to FASN expression in MyLa2059. (**D**) Immunohistochemistry on skin biopsies verified the presence of FASN in MF patients (Figure 1D upper right, lower right, and left). Healthy skin revealed positivity only in skin adnexa (Figure 1D upper left). In upper right, positive tumour cells in a Pautrier’s microabscess are depicted. In the lower row, MF samples with strong and weak to moderate positivity are shown. Individual bars indicate image scale, thus upper left bar: 500 μm, upper right bar: 100 μm, and lower left and right bar: 200 μm, respectively. Statical significance between the control (healthy PMBCs) and individual samples were tested with two-sided Student’s *t* test. Asterisks above each column in (**B**) indicate the significance of difference compared to the control (* *p* ≤ 0.05; ** *p* ≤ 0.01).

**Figure 2 cancers-14-04491-f002:**
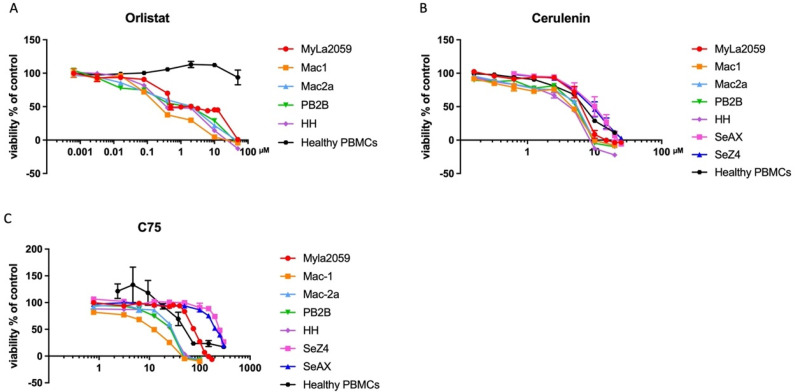
Inhibition of FASN reduces the viability of CTCL cell lines. CTCL cell lines and PBMCs from healthy donors were treated with three different FASN inhibitors; (**A**) Orlistat, (**B**) Cerulenin, and (**C**) C75. After incubation for 48 h, cell viability was analysed by MTT assay as described in material and methods. All the CTCL cell lines showed similar sensitivity towards Orlistat, Cerulenin, and C75, but Orlistat did not affect healthy PBMCs. The concentration ranges for Orlistat were: 0.00064 μM–50 μM, Cerulenin: 0.16 μM–25 μM, and C75: 0.78 μM–300 μM, respectively.

**Figure 3 cancers-14-04491-f003:**
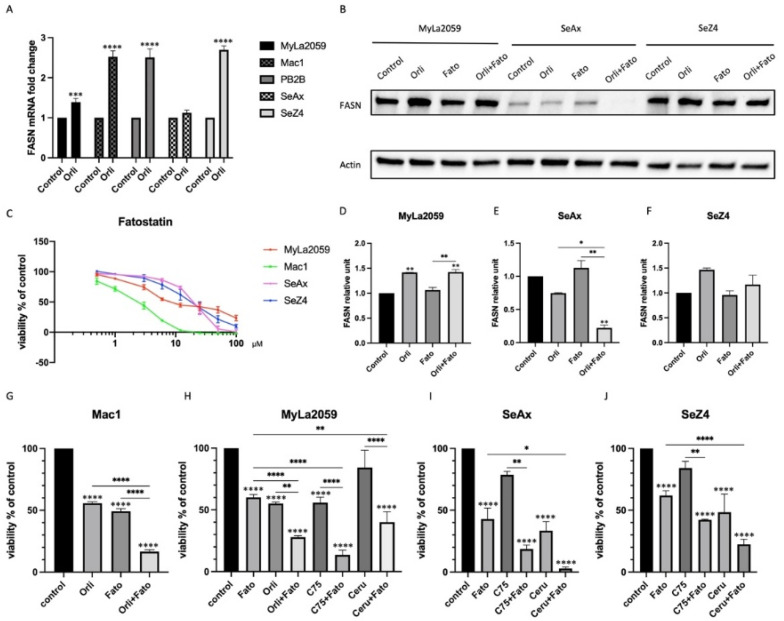
Inhibition of SREBP could prevent the upregulation of FASN induced by Orlistat or cause further inhibition on FASN, enhancing the potency of Orlistat on cell viability. (**A**) Orlistat caused the increase of FASN mRNA and (**B**) FASN protein, except in SeAx cells. Inhibition of SREBP by Fatostatin partially counteracted this upregulation in SeZ4, while it enhanced the inhibition of Orlistat on FASN in SeAx cells. (**D**–**F**) Quantification of protein level in western blot. (**C**) Fatostatin also impaired the cell viability in CTCL cell lines, but (**G**) when combined with the FASN inhibitor, it could incur more suppression compared to each of these inhibitors alone. Unless otherwise stated, the concentration of Cerulenin, C75, and Fatostatin used in this study was, respectively, 6 µM, 75 µM, and 5 µM in Mac1 and MyLa2059 and 10 µM, 150 µM, and 20 µM in SeAx and SeZ4. The concentration of Orlistat used in SeAx and SeZ4 was 75 µM, while in Mac1 and MyLa2059 it was 1 µM for qPCR and MTT assay, and 10 µM for western blot. Western blot and MTT assay were performed after 48 h of stimulation, and qPCR was performed after 24 h. Statical significance between the control and individual samples were tested with two-sided Student’s *t* test for graph A. One-way ANOVA (Tukey’s) test was used for graph (**D**–**J**) as described in materials and methods. Asterisks above each column in (**A**,**D**–**J**) indicate the significance of difference compared to the control, while bars with asterisks show the statistical significance between one inhibitor alone and two inhibitors in combination (* *p* ≤ 0.05; ** *p* ≤ 0.01; *** *p* ≤ 0.001, **** *p* ≤ 0.0001).

**Figure 4 cancers-14-04491-f004:**
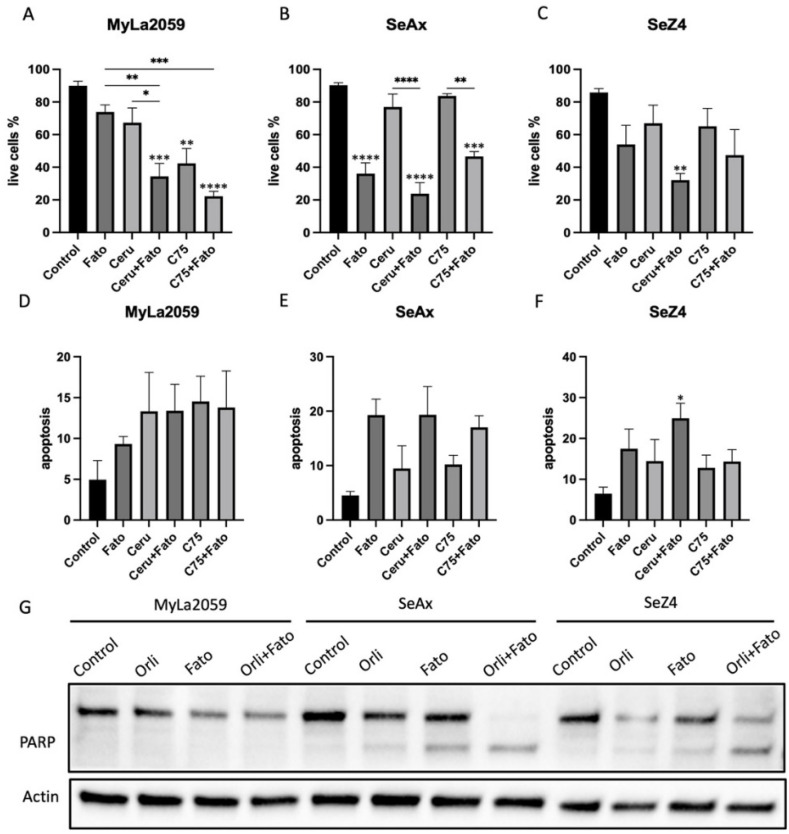
SREBP and FASN inhibitors attenuated cell survival and induced apoptosis in CTCL cell lines. (**A**–**C**) Cell survival analysed by flowcytometry showed damaged cell survival by Fatostatin, Cerulenin, and C75 in MyLa2059, SeAx, and SeZ4, with more effects in the combination group. (**D**–**F**) Apoptosis evaluated by both flowcytometry and (**G**) western blot exhibited similar results. Statical significance was tested with one-way ANOVA with Tukey´s multiple comparisons, as described in material and methods. Asterisks above each column in (**A**–**F**) indicate the significance of difference compared to the control, while bars with asterisks mark the significance between an inhibitor and any combination containing that inhibitor (* *p* ≤ 0.05; ** *p* ≤ 0.01; *** *p* ≤ 0.001; **** *p* ≤ 0.0001).

**Figure 5 cancers-14-04491-f005:**
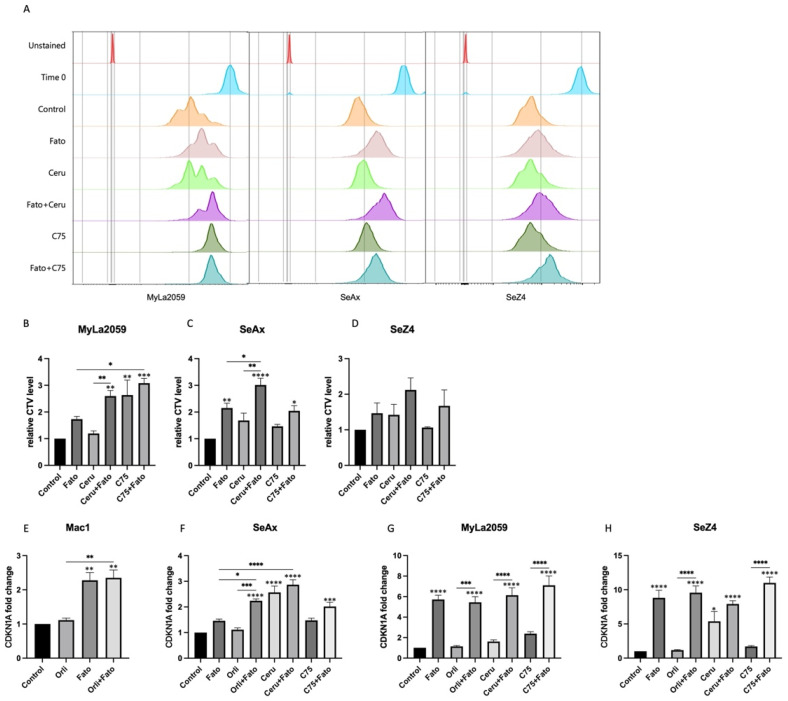
SREBP and FASN inhibitors impair cell proliferation and increase CDKN1A expression in CTCL cell lines. (**A**) Representative CellTrace violet (CTV) fluorescence intensity profiles of the indicated CTCL cell lines treated with the given inhibitors, and (**B**–**D**) illustrate the mean fluorescence intensity (MFI). (**C**–**H**) depicted the relative expression of CDKN1A in three CTCL cell lines treated with the indicated SREBP and FASN inhibitors. The expression levels were normalized to CDKN1A expression in untreated cells. Statical significance was calculated with a one-way ANOVA test with Tukey´s multiple comparison, as described in materials and methods. Asterisks above each column in (**B**–**H**) indicate the significance of difference compared to the control. Bars with asterisks signify the difference between an inhibitor and any combination with this inhibitor (* *p* ≤ 0.05; ** *p* ≤ 0.01; *** *p* ≤ 0.001; **** *p* ≤ 0.0001).

**Figure 6 cancers-14-04491-f006:**
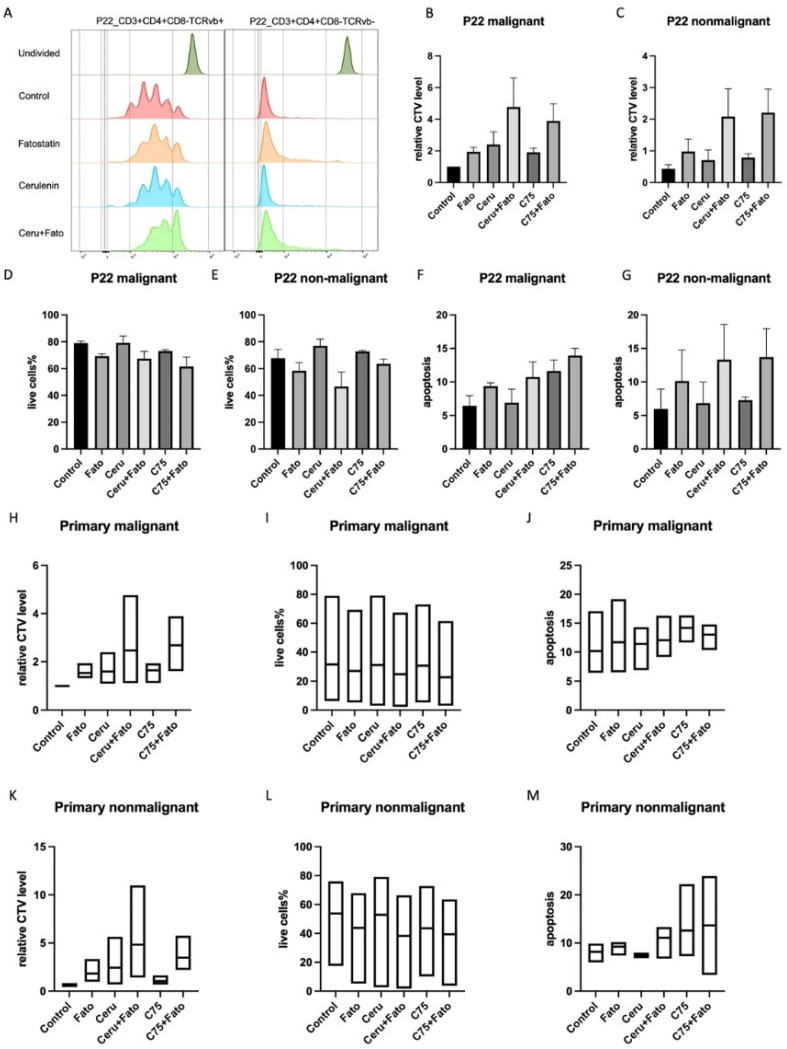
The effects of FASN and SREBP inhibitors in primary cells. (**A**–**G**) reveal the effects of FASN and SREBP inhibitors on patient 22. (**H**–**M**) are the collective data of patient P22, P6, and P16. Statical significance testing was performed with a one-way ANOVA (Tukey´s multiple comparison) test described in materials and methods.

**Table 1 cancers-14-04491-t001:** Numbers of MF patients according to their clinical stage and frequency of FASN positive lymphocytes in the biopsies.

	FASN Positive Lymphocytes			
Clinical Stage	Negative	1–10%	11–49%	>50%
IA	1	2		
IB		5	4	
IIA			2	
IIB		3	1	1

## Data Availability

The data presented in this study are available in the manuscript and in the Appendix A.

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
