# Peer review of "Concomitant Inhibition of FASN and SREBP Provides a Promising Therapy for CTCL"

_cancers, 2022, doi:10.3390/cancers14184491_

Round 1

Reviewer 1 Report

In the present study the authors have shown that FASN is highly expressed in CTCL cell lines and in peripheral blood mono-nuclear cells (PBMCs) from CTCL patients and thus, inhibition of FASN was found to impair cell viability, and proliferation. Additionally, the authors showed that combination of FASN and SREBP inhibitors enhanced the cytotoxic effects on both CTCL cell lines and PBMCs from SS patients, with valid inhibition on cell proliferation. Essentially, the authors found that compared to non-malignant cells, primary malignant cells are more sensitive to the inhibition of FASN and SREBP, making the combination treatment a promising novel therapeutic strategy in CTCL.

Need major revision on the following points.

1.     The figure legends are not described properly with proper alphabetical numbering. Discuss.

2.     The labelling of the figures doesn’t at all matches with that of the result section, which makes it very difficult to understand. Correction is instantly needed.

3.     Describe the inhibitors used and their mode of action before directly moving on to the result section.

4.     Clearly discuss the clinical significance and the outcome of the study done. 

5.     The result section for fig 1A describe that all the CTCL cell lines had increased FASN expression as compared to healthy PBMC at both the protein and mRNA level. But healthy PBMC panel was missing in the western. Include the protein expression analysis for FASN in 2 or 3 samples of healthy PBMCs to strengthen the data.

6.     Provide image scale for fig 1C IHC studies.

7.     Provide IC50 values for fig2.

8.  Why normal PBMCs sample are not included in fig 2C for C75 treatment? Include the same.

9.     How will the author describe the increase in FASN expression at RNA and protein levels after treatment with Orlistat? Is there any background reference for such observation? Why didn’t the author included healthy PBMCs to access FASN levels after treatment with Orlistat? Will there be a difference in the results obtained between CTCLs and healthy PBMCs?

10.  Why there is reduced PARP levels in the combination treatment group in fig 4C? The authors should check for cleaved caspase 3 levels as it directly relates to apoptosis induction.

11.  The authors must explain why they suddenly checked CDKN1A levels? A protein level estimation for the CDKN1A was required to conclude the data. Does the authors perform any experiments for the cell cycle analysis to relate to CDKN1A levels.

12.  The fig 6 part of the result section is too confusing. Please rewrite the part describing the patient samples involved.

13.  The strength of the work can be improved by silencing the SREBP1 in the CTCL cell lines and analyzing the cytotoxic effect upon treatment with FASN inhibitors compared to SREBP1 expressing cells. The authors should perform cell viability assay, apoptosis assay, western analysis of cleaved caspase 3 under the same condition.

Author Response

In the present study the authors have shown that FASN is highly expressed in CTCL cell lines and in peripheral blood mono-nuclear cells (PBMCs) from CTCL patients and thus, inhibition of FASN was found to impair cell viability, and proliferation. Additionally, the authors showed that combination of FASN and SREBP inhibitors enhanced the cytotoxic effects on both CTCL cell lines and PBMCs from SS patients, with valid inhibition on cell proliferation. Essentially, the authors found that compared to non-malignant cells, primary malignant cells are more sensitive to the inhibition of FASN and SREBP, making the combination treatment a promising novel therapeutic strategy in CTCL.

Need major revision on the following points.

  1. The figure legends are not described properly with proper alphabetical numbering. Discuss.

Thanks for pointing this out. We have updated all figures.

  1. The labelling of the figures doesn’t at all matches with that of the result section, which makes it very difficult to understand. Correction is instantly needed.

Thanks for pointing this out. We apologize for this and have now updated all figures.

  1. Describe the inhibitors used and their mode of action before directly moving on to the result section.

We have added a section describing the MOA of the inhibitors in the introduction.

  1. Clearly discuss the clinical significance and the outcome of the study done.

We have included additional discussion of the clinical significance in the combination of inhibitors of both FASN and SREBP.

  1. The result section for fig 1A describe that all the CTCL cell lines had increased FASN expression as compared to healthy PBMC at both the protein and mRNA level. But healthy PBMC panel was missing in the western. Include the protein expression analysis for FASN in 2 or 3 samples of healthy PBMCs to strengthen the data.

We have now performed analysis of FASN protein on PBMCs from three individual healthy donors. Data is now included in figure 1 (fig 1C).

  1. Provide image scale for fig 1C IHC studies.

Thanks for pointing this out. We have added image scales for each IHC picture and added description of this in the figure legend (fig1).

  1. Provide IC50 values for fig2.

We have determined the IC50 values for each inhibitor towards each cell line (and PBMCs from healthy donors). We have collected the result in a supplementary table (supplementary table 1).

  1. Why normal PBMCs sample are not included in fig 2C for C75 treatment? Include the same.

We have now added cell viability data from treatment of PBMCs from three individual healthy donors treated with C75 for 48 hours.

  1. How will the author describe the increase in FASN expression at RNA and protein levels after treatment with Orlistat? Is there any background reference for such observation? Why didn’t the author included healthy PBMCs to access FASN levels after treatment with Orlistat? Will there be a difference in the results obtained between CTCLs and healthy PBMCs?

This is an important question. It is known that reduced levels of free fatty acids induced increased expression of FASN through induction of SREBP mediated transcription of the FASN gene as described in the introduction. This is what lead us to investigate whether the observed increase of FASN mRNA and protein expression (in some, but not all cells). We have now included analysis (qPCR and Western blotting) of FASN expression in PBMCs from three individual heathy donors treated with Orlistat (supplementary fig 1). As shown in supp. Fig 1, we find no expression of FASN protein in healthy PBMCs, and neither that Orlistat induces the expression of FASN mRNA or protein.

  1. Why there is reduced PARP levels in the combination treatment group in fig 4C? The authors should check for cleaved caspase 3 levels as it directly relates to apoptosis induction.

This is a very good question to which we have no good answer. We have tried to analyze caspase 3 expression by Western blot but have unfortunately not achieved useful data. Instead, we have analyzed apoptosis by flow cytometry (annexin/PI staining).

  1. The authors must explain why they suddenly checked CDKN1A levels? A protein level estimation for the CDKN1A was required to conclude the data. Does the authors perform any experiments for the cell cycle analysis to relate to CDKN1A levels.

This is a good point. Induction of CDKN1A following inhibition of FASN has been previously described in connection to FASN-inhibitor mediated suppression of proliferation. This is now descripted in the result section and the reference in now included.

  1. The fig 6 part of the result section is too confusing. Please rewrite the part describing the patient samples involved.

The text have been updated. We hope it is more clear now.

  1. The strength of the work can be improved by silencing the SREBP1 in the CTCL cell lines and analyzing the cytotoxic effect upon treatment with FASN inhibitors compared to SREBP1 expressing cells. The authors should perform cell viability assay, apoptosis assay, western analysis of cleaved caspase 3 under the same condition.

This is an excellent suggestion. As we see induction of both SREBP1 and SREBP2 following inhibition of FASN enzymatic activity, it is likely that silencing of both is needed. This is not an easy experiment to set up, and unfortunately, due to the time limitations for this revision we have not been able to perform this experiment.

Reviewer 2 Report

The MS entitled "Concomitant inhibition of FASN and SREBP provides a promising therapy for CTCL” is well written and presents a substantial amount of very interesting data.

1.      It is better to use abbreviation list in MS.

2.      Please check key words based on mesh term.

3.      Fig 3: explain in more detail.

Author Response

The MS entitled "Concomitant inhibition of FASN and SREBP provides a promising therapy for CTCL” is well written and presents a substantial amount of very interesting data.

  1. It is better to use abbreviation list in MS.

Thank you for the suggestion. If this is possible to do so in Cancers, we will prepare an abbreviation list.

  1. Please check key words based on mesh term.

Thank you for pointing this out. We will change the mesh terms if possible.

  1. Fig 3: explain in more detail.

We have updated the text for figure 3 and added additional supplementary data (supplementary figure 1). We hope this has improved the explanation of results shown in figure 3.

Reviewer 3 Report

The authors studied the role of FASNin CTCL and found that FASN inhibitor impairs cell viability, survival and proliferation and FASN inhibitor upregulates FASN expression and combination of FASN inhibition and SREBP inhibiton enhanced the effects.

Figures in general

Please add explanation about * p < 0.05 etc. Please describe tests (student t test etc) in figure legends.

For the multiple comparison, student’s t test is not good. Please test with one-way ANOVA (Dunnet, or Tukey)

Figure 3

Fatostatin, Orlistat, these name abbreviation (Fato, Orli) should be described in figure legend.

In the figure, there are no Fig 3E-F. Label needs to be corrected in the figure and manuscripts.

Figure 5

Labels in Fig 5A are too small. Please make it bigger.

Figure 6

Labels are missing. Please correct it.

Page 5 160

The authors showed that FASN inhibition upregulates FASN expression and addition of SREBP inhibitor will increase the effect of FASN inhibition by decreasing FASN expression.

The authors cited a paper and described that loss of FASN can promote activation of SREBP2. So can you show that inhibition of FASN by Orlistat activates SREBP2?

Page 9 212

When you define malignant cells from PBMC in Sezary patients, did you also confirm CD7 negativity? If possible, it is good to confirm CD7 negativity in malignant cells from Sezary patients. 

Author Response

The authors studied the role of FASNin CTCL and found that FASN inhibitor impairs cell viability, survival and proliferation and FASN inhibitor upregulates FASN expression and combination of FASN inhibition and SREBP inhibiton enhanced the effects.

Figures in general

Please add explanation about * p < 0.05 etc. Please describe tests (student t test etc) in figure legends.

Thank you for pointing this out. We have added an explanation in materials and methods and have also described the tests in the figure legends.

For the multiple comparison, student’s t test is not good. Please test with one-way ANOVA (Dunnet, or Tukey)

We have not made any multiple comparisons.

Figure 3

Fatostatin, Orlistat, these name abbreviation (Fato, Orli) should be described in figure legend.

In the figure, there are no Fig 3E-F. Label needs to be corrected in the figure and manuscripts.

Thanks for pointing this out. We have updated all figures.

Figure 5

Labels in Fig 5A are too small. Please make it bigger.

We agree and apologizes for this. We have now changes the size of Fig 5A, and hope labels can be read now.

Figure 6

Labels are missing. Please correct it.

Thanks for pointing this out. We have updated all figures.

The authors showed that FASN inhibition upregulates FASN expression and addition of SREBP inhibitor will increase the effect of FASN inhibition by decreasing FASN expression.

The authors cited a paper and described that loss of FASN can promote activation of SREBP2. So can you show that inhibition of FASN by Orlistat activates SREBP2?

Thanks for pointing this important point out. We have included data showing that Orlistat result in increased expression of both SREBP1 and SREBP2 (supplementary fig 4). Activity of either SREPB1 or SREBP2 result in increased expression of FASN, so we believe that this is what our data shows. 

When you define malignant cells from PBMC in Sezary patients, did you also confirm CD7 negativity? If possible, it is good to confirm CD7 negativity in malignant cells from Sezary patients. 

We have not used CD7 to define the malignant clone. Instead, we have identified the malignant clone based on analysis of TCRvb expression. We have now added a reference describing how we use analysis of TCRvb to identify the malignant clone in SS patients.

Reviewer 4 Report

This study showed FASN is highly expressed din CTCL cell lines and PBMCs from the Sezary syndrome patients and CTCL patients. In contrast, FASN expression was not detected in healthy control. The authors also demonstrated the inhibition of CTCL cell proliferation by FASN inhibitors. Overall, it is a sound research unrevealing interesting and novel findings.

Author Response

This study showed FASN is highly expressed din CTCL cell lines and PBMCs from the Sezary syndrome patients and CTCL patients. In contrast, FASN expression was not detected in healthy control. The authors also demonstrated the inhibition of CTCL cell proliferation by FASN inhibitors. Overall, it is a sound research unrevealing interesting and novel findings.

We thank the reviewer for the kind words.

Round 2

Reviewer 3 Report

The authors answered properly other than my comments about multiple comparison test.

For example, figure 3D, E, F, G, H, I J, each figure has more than three groups and we can compare every two pair of groups. In this situation, multiple comparison test is better because if you increase the number of groups, it will increase the chance of getting significant change. Please check what statistical test is proper for each comparison.

Author Response

Dear reviewer,

Thank you for the suggestion. We have now performed multiple comparison statical analysis (Tukey's) for all relevant figures inc supplementary figures and table, and all figures has been updated.